# Digital Trends in the Italian Beer Market: A Time-Series and Search Engine Optimisation Analysis of Gluten-Free and Low/No-Alcohol Beers

**DOI:** 10.3390/foods14213789

**Published:** 2025-11-05

**Authors:** Pietro Chinnici, Katya Carbone, Francesco Licciardo

**Affiliations:** 1Department of Agricultural, Food and Forestry Sciences, Università Degli Studi di Palermo, 90123 Palermo, Italy; pietro.chinnici01@unipa.it; 2Research Centre for Olive, Fruit and Citrus Crops, CREA, Via di Fioranello 52, 00134 Rome, Italy; katya.carbone@crea.gov.it; 3Research Centre for Agricultural Policies and Bio-Economy, CREA, Via Barberini 36, 00187 Rome, Italy

**Keywords:** non-alcoholic and low-alcoholic beers, gluten-free beers, search engine optimisation analysis, google trends, time series, forecast analysis, seasonal analysis

## Abstract

Beer consumption patterns are evolving, with gluten-free beers (GFBs) and low- and no-alcohol beers (NABLABs) exhibiting continuous growth, underpinned by health-conscious consumers, younger generations’ preference for moderate drinking, and a rising awareness of gluten intolerance. This study investigates whether online search behaviour reflects these market changes and anticipates future consumption trends. A combined methodological framework was applied, integrating time-series analysis of Google Trends data—based on a decomposition model with a five-year forecast—with descriptive and semantic insights from Search Engine Optimisation (SEO) techniques using the specialised SEOZoom platform. The statistical decomposition enabled the identification of long-term trends, cyclical variations, and seasonal patterns in user interest. Italy was selected as a case study, representing a market where these niche segments have grown considerably despite a traditionally limited craft beer culture. The results reveal a steady rise in online interest in both GFB and NABLAB; GFB searches are primarily linked to health and dietary concerns, whilst NABLAB queries encompass a broader set of topics, including physiology, legislation, and consumption contexts. The forecasts confirm the persistence of this positive trend over the next five years. The approach demonstrates the potential of integrating digital and statistical tools to monitor emerging consumption dynamics and guide strategic decisions in the beverage sector.

## 1. Introduction

The concept of functional foods has gained significant attention in recent years, bridging the gap between traditional nutrition and emerging health-conscious consumer trends. These foods, formulated to deliver specific health benefits beyond basic nutrition, represent a growing segment in the global food market [1]. As consumers become more aware of the connection between diet and health, functional foods present a promising opportunity to address a range of health concerns through everyday dietary choices [2].

The growth in the popularity of these foods is indicative of a more general shift in consumer attitudes towards health and wellness [3,4]. Modern consumers, in fact, are no longer merely seeking sustenance; they are also looking for foods that can contribute positively to their overall well-being. This trend has resulted in a significant increase in research and development in the food and beverage industry, with a focus on the creation of products that offer targeted health benefits while maintaining a high level of sensory experience. In this context, the emergence of low-alcohol, non-alcoholic (NABLAB) and gluten-free (GFB) beers is unsurprising [5,6,7,8]. These special beers accounted for 2.1% of total Italian beer consumption in 2024, representing an increase of 13.4% in comparison with the previous year [9].

The craft beer sector in Italy has expanded considerably over the past two decades, developing from a niche pursuit among enthusiasts into a phenomenon of both economic and cultural importance [10,11]. Domestic production increased from 13.5 million hectolitres in 2014 to 17.2 million in 2024, while consumption followed a comparable trajectory, reaching 21.5 million hectolitres in 2024, representing a 21% rise compared with 2014. Per capita beer consumption is currently estimated at 37 L and appears to have stabilised, with no substantial further growth observed. In this context, competitive pressures are emerging, as market fragmentation and consolidation among major players are making it increasingly difficult for small businesses, such as microbreweries, to survive. Following a period of considerable growth, in fact, these businesses are now encountering a phase of stagnation [12], with the bulk of production increasingly concentrated among large-scale breweries [13]. As a consequence, several studies are currently underway exploring the development of innovative beers with the objective of attracting consumers and ensuring market penetration [14,15,16].

However, the focus of research has not been limited to product innovation. Increasing attention has also been paid to communication strategies. In countries such as Spain and Italy, research has examined the potential influence of sustainability-driven branding initiatives on consumer perceptions, with a view to enhancing the commercial performance of microbreweries [17,18]. Today, the organoleptic quality of beer alone is no longer sufficient to ensure the commercial success or survival of small-scale producers. The ability to communicate added value in a way that is accessible to a wide audience has become essential [19]. Accordingly, the adoption of strategic marketing approaches is increasingly necessary for the economic sustainability of craft breweries [20,21]. Moreover, the deployment of digital tools designed to engage specialised and niche consumer segments represents a critical component of business sustainability for microbreweries [22,23].

Among these tools, Search Engine Optimisation (SEO) plays a pivotal role. Enhancing online visibility through the strategic use of keywords, relevant content, and optimised technical infrastructure enables breweries to improve their positioning on search engines, capture latent demand, and influence consumer purchasing behaviour [24]. The analysis of online search trends, supported by SEO software and platforms such as Google Trends (GT) and SEOZoom, offers new opportunities to understand consumer needs and inform business decisions, particularly in areas such as product development, branding, and distribution strategies.

Studying the trend of online searches related to GFB and NABLAB is strategically important because search engine queries represent timely behavioural traces of consumer needs, curiosities, and intentions. Unlike sales data, which captures choices already made, search volumes and dynamics anticipate shifts in demand, intercepting exploratory and evaluative phases (information, comparison, availability, point of sale search). In this sense, online search volumes with seasonal patterns and SEO analysis offer leading indicators of changes in beer consumption styles, which are increasingly influenced by health and wellness issues (celiac disease, gluten sensitivity, alcohol moderation), new occasions for consumption, and greater attention to perceived quality and transparency.

Despite the importance of these digital strategies, few studies have examined how search-engine data can be systematically applied to validate and extend traditional market research in the beer sector. Existing research has primarily focused either on product innovation or on communication strategies in general, but less attention has been devoted to the integration of digital behavioural data into the analysis of emerging categories such as GFB and NABLAB. Moreover, while SEO tools like SEOZoom provide a descriptive picture of search intent and semantic structures, they rarely account for the temporal dynamics of consumer interest. Conversely, time-series analyses from platforms like GT capture long-run patterns and seasonality, but lack semantic depth. This disconnect leaves a gap in understanding the interplay between what consumers search for and when they do so.

The research contributes to the existing literature on the Italian beer market by introducing a novel methodological approach that leverages digital consumer behaviour data. This approach not only validates observed market dynamics but also offers predictive insights into future trends, potentially supporting strategic decision-making in the industry.

In light of these considerations, this study investigates online search behaviour in Italy relating to selected speciality beers, namely gluten-free and low- or non-alcoholic beers. Specifically, it addresses the following research questions:In relation to observed market changes concerning GFB and NABLAB, can the evolution of beer consumption be confirmed through the online search activity of consumers?What new patterns of consumer interest and intent are emerging in the Italian beer market, and what future directions are likely to shape demand?How can the integration of time-series analysis and descriptive SEO analysis offer a more robust framework for understanding consumer behaviour in niche food and beverage markets?

## 2. Data and Methods

The methodological approach adopted is of a descriptive nature, with GT being the primary instrument utilised for the evaluation of users’ online search behaviour in relation to GFB and NABLAB. Statistical techniques were applied on GT data to identify recurring temporal patterns and systematic seasonal variations. Finally, an SEO perspective was incorporated by combining GT with specialised software. A synopsis of the approach used is presented in Figure 1.

Specifically, the initial phase of the analysis entailed the identification of search-relevant keywords that reflect online user interest trends, with a particular focus on search volume and temporal dynamics. To achieve this objective, the GT tool was utilised. This resource, provided by Google, is widely used for the purpose of monitoring the popularity of search terms over time and across geographical regions [25,26]. To facilitate a more rigorous examination of the GT dataset, a time series analysis was conducted to forecast search interest over a five-year period. The historical series was subjected to decomposition to identify its constituent components (trend, seasonal, and irregular). This process enabled the identification of recurrent temporal patterns and systematic seasonal fluctuations. This decomposition further enabled the generation of forward projections of search volumes, providing an estimate of expected future dynamics conditional on the observed historical structure. The time series decomposition approach adopted in this study aligns with a well-established body of applied literature, which demonstrates its efficacy in forecasting demand. Recent studies demonstrate that decomposition provides enhanced predictive efficacy in comparison to alternative classical models for products characterised by well-defined seasonality [27]. Furthermore, it has been shown that decomposition can enhance the comprehensibility and accuracy of predictive models, even in more complex contexts [28].

The second phase of the analysis evaluated the digital visibility and search engine ranking performance of the selected keywords. SEO is a series of practices aimed at improving a website’s visibility in organic search engine results, thereby substantially increasing web traffic and potential consumer engagement. The significance of SEO analysis in the context of effective corporate communication has been increasingly recognised [29,30]. This component of the analysis was conducted using SEOzoom. The software provides comprehensive SEO analytics and facilitates the examination of keyword performance across a database indexing more than two billion Italian-language web pages [31,32,33].

### 2.1. Google Trends Tool

Google’s free, publicly accessible tool, known as GT, was developed for the purpose of analysing the popularity of search keywords over time and across various regions [34]. Cebrián and Domenech [35] underline that the utilisation of GT in scientific literature has grown in popularity, as its reports offer a means to gauge the population’s interest in diverse subjects, including medicine, economics, politics, and more. Among non-traditional data sources, GT has been identified as a prominent instrument within the empirical economic literature. This recognition can be attributed to its ability to capture real-time information on public interest and behaviour [36,37,38,39].

As highlighted in [40], Google Search provides an excellent platform for the observation of consumers’ information search activities, providing an immediate reflection of the needs, desires, demands and interests of its users. It provides normalised data that reflects the frequency of specific keyword searches relative to the total number of searches conducted within a selected timeframe and location [26,41,42]. This normalisation renders it especially efficacious in the identification of temporal patterns and regional variations in the public interest. The data were derived from a representative sample of user-search activities that was representative and anonymised. The primary output of the model is a time-series dataset, in which the values are scaled from zero to 100, with 100 representing the peak search volume [40]. Although the tool does not disclose absolute search volumes, its relative scale allows for meaningful comparisons between different search terms, periods, or regions.

### 2.2. Time Series Analysis

The analysis of time series using data obtained from GT is well established in the literature and has been applied across various fields of research [43,44,45]. In this study, search volume data from GT were used on a monthly basis for the keywords GFB and NABLAB. The analysis covered the period from 1 January 2004 to 4 April 2025, with a focus on the Italian market. Furthermore, a five-year forecasting analysis was conducted to identify future trends in user searches.

Preliminary analyses were conducted to ascertain the most appropriate model. Following several approaches [46,47], an autocorrelation function (ACF) analysis was conducted to evaluate the existence of recurrent patterns in the historical data that might have a bearing on the selection of the model. As is well established, ACF is a statistical tool that measures the degree of correlation between time series values and their lagged counterparts over a defined number of periods [48]. As the data was collected on a monthly basis, the analysis was conducted with a 12-month lag in order to capture potential annual seasonal patterns. In this context, the term ‘lag’ refers to the number of time periods utilised to compare the series with itself. Specifically, a lag k denotes the time distance of k units (e.g., days, months, or years) between observations. The correlation values were then compared with 95% confidence intervals in order to determine the statistical significance of the observed peaks [48]. The results indicated that the autocorrelation values exceeded the confidence intervals across all lags considered, thereby highlighting a strong and persistent seasonal correlation (see Appendix A).

The presence of seasonality necessitated the adoption of a decomposition approach for time series analysis. This modelling strategy isolates and separately estimates trend, seasonal, and irregular components, thereby enhancing both interpretability and analytical accuracy. In accordance with the findings of several authors [49,50], a multiplicative decomposition model was applied, given that the seasonal variation increased in magnitude over time, an expected characteristic of processes in which fluctuations are proportional to the average level of the series. Each time series was treated as a discrete stochastic process with equally spaced monthly observations and annual periodicity.

To assess the accuracy and robustness of the forecasting models, three standard error metrics were calculated: Mean Absolute Percentage Error (MAPE), Mean Absolute Deviation (MAD), and Mean Squared Deviation (MSD). MAPE quantifies the average absolute error in percentage terms, thus providing a unit-free measure of predictive accuracy; lower MAPE values indicate superior performance. MAD expresses the average absolute error in the units of the observed variable, offering a direct measure of deviation from the actual values. The MSD, calculated as the mean of the squared deviations, places greater weight on larger errors and facilitates the detection of anomalies or outliers. The combined application of these metrics allows for a comprehensive evaluation of the model performance, as each offers a distinct perspective on predictive validity [48,51].

Finally, a five-year forecast was generated to estimate future trends in search volumes. This forecast employed the seasonal decomposition model identified during the analysis phase, combined with projections derived from the extrapolation of the trend components. Consistent with the methodological recommendations [51], this approach separates seasonal fluctuations from long-term trends, thereby enhancing the projection accuracy. A five-year horizon was selected to strike a balance between the provision of actionable insights for strategic planning and the reduction in uncertainty inherent in long-term forecasts.

All analyses were conducted using Minitab software (version 19).

### 2.3. SEOzoom Tool

The evolving digital behaviour of consumers highlights the importance of search engine optimisation (SEO) in business communication. Research shows that SEO strategies enhance organisational authority and content relevance, attributes prioritised by search engines such as Google and Bing [52,53]. Effective SEO improves discoverability and influences perceptions of credibility and authority, factors linked to purchasing behaviour [29,54].

Integrating SEO into business communication pedagogy equips students with essential web writing skills for achieving visibility in search engine results pages (SERPs) and contributes to competencies in audience analysis, digital rhetoric, and keyword research [55]. Strategic keyword selection, supported by research tools, aligns content with consumer search intentions across different stages of the customer’s journey [56]. Well-designed SEO strategies also improve the quality of organic links, benefiting both consumers and advertisers [29,57].

Therefore, choosing the most suitable SEO analysis software is crucial. In this study, SEOzoom was used. Specifically, SEOZoom is a software platform for analysing the Italian digital market. It indexes over two billion Italian-language web pages and provides real-time monitoring of keyword positions and SERP metrics [32,33]. The core functions include keyword trend analysis, domain benchmarking, visibility measurement, and competitive profiling. The software reports search volume, difficulty, seasonality, and cost per click, supporting demand forecasting in cyclical sectors such as tourism and retail. Its SERP analysis identifies ranking pages, backlink profiles, and domain authority, whereas the SEOZoom Visibility Index estimates long-term organic reach. A semantic mapping tool further classifies related searches into Queries, Prepositions, and Actions, offering insights into user intent and content optimisation opportunities.

SEOZoom facilitates the extraction and interpretation of key metrics describing both user search behaviour and online content visibility. The indicators selected and analysed for this study are presented in Table 1.

The integration of the indicators presented in Table 1 offered a comprehensive understanding of how GFB and NABLAB are represented, searched for, and promoted within the Italian digital environment. SEOZoom has therefore provided a solid interpretative framework for linking online search dynamics to potential market trends.

The analysis covered the period from 1 January 2024 to 31 December 2024 and was restricted to the Italian market.

## 3. Results

The analysis was conducted on web users in Italy (potential beer consumers); accordingly, all data pertained to the national context. It is important to note that this research was based on Italian keywords, which have been translated for the sake of clarity.

In the initial phase of the analysis, following several trials using GT, two specific keywords were selected: GFB and NABLAB. These terms were identified because of their relevance to evolving consumer preferences within the beverage sector and their discernible presence in online search behaviour. Time series and SEOzoom analyses were subsequently performed for the selected keywords. The outcomes of both analyses for the aforementioned keywords are presented below.

The data derived from the semantic analysis, visualised through the graphs generated by SEOzoom, were presented in Italian, reflecting the fact that the analysis targeted Italian consumers. To facilitate the comprehension of the graphs in Appendix B, accompanying tables containing the translated keywords are provided.

### 3.1. Time Series Analysis for Gluten-Free Beers

Figure 2 presents the decomposition of the historical series, which shows a clear long-term positive trend. The observed values for GFB (blue line) began to increase only after 2011, coinciding with the first recorded web searches, and subsequently entered a phase of moderate growth, peaking between 2018 and 2022. The trend component (green line) evolved almost linearly, indicating structural growth in search volumes. This trend is accompanied by a distinct seasonal component and a relatively low residual variability. Forecast projections (purple triangles) suggest continued growth until 2030, with an amplification of the seasonal effect, which is reflected in more pronounced peaks and troughs.

The fitted curve (red line) closely followed the observed values, demonstrating good predictive accuracy. Model performance metrics confirm this: the 14.72% MAPE indicates a limited average percentage error, well within the <20% threshold generally regarded as satisfactory for series with long-term trends and seasonality. The MAD of 5.87 implies that forecasts deviate by approximately six units from observed search volumes, while the MSD of 71.06 corresponds to a root mean square error of 8.43 units. Considering that Google Trends scales data from zero to 100, these values confirm adequate predictive capability despite some unexplained variability. Overall, the error measures indicated that the model robustly captured both the long-term trends and seasonal fluctuations.

The results suggest that the demand for GFB is characterised by structural long-term growth combined with intensifying seasonality. The strong agreement between the observed and estimated values, together with the model’s capacity to project plausible future dynamics, supports its suitability for forecasting online search volumes in this context.

Seasonal decomposition using a multiplicative model (Figure 3) confirmed a marked cyclical component in the monthly searches. Seasonal indices revealed the highest values during summer, peaking in August (index 1.390), approximately 10% above the annual average. Conversely, the lowest indices occurred in the winter months, particularly November and March, reflecting reduced interest. The Detrended Data by Season confirm that the residual variability is concentrated in the summer, consistent with consumption patterns and favourable weather conditions. Residual analysis showed no systematic errors, indicating the correct identification of the seasonal component. The multiplicative model, which assumes proportionality between seasonal amplitude and series level, is particularly appropriate given the observed percentage-based fluctuations.

In summary, the analysis demonstrates that the demand for GFB follows a strongly seasonal pattern, with consumption peaking in summer. These findings have practical implications for production, logistics, and promotional strategies in this sector.

### 3.2. SEOzoom Analysis for Gluten-Free Beer

The analysis of monthly variation in indexed webpages revealed marked fluctuations in content volume associated with the keyword GFB, ranging from approximately 7.15 to 8.14 million pages during the observation period (Figure 4). The series is characterised by an initial decline (7.67 to 7.15 million), followed by recovery and stabilisation between 7.7 and 7.9 million, and a final increase, peaking at 8.14 million indexed pages. Although the period examined covered only one year, the results confirmed the upward trend in the number of web pages indexed on the search engine (see Figure 4).

The seasonal dynamics of user interest, measured by the average monthly search volume, are shown in Figure 5. Although not a direct indicator of purchasing behaviour, keyword searches provide valuable information on consumer awareness, informational needs, and receptiveness to product-related content. The data highlight a pronounced seasonal pattern: peak volumes of approximately 5400 searches occur between May and August, corresponding to the Italian summer, while minimum levels (2400) are recorded in January and February. This cyclical behaviour suggests that online interest in GFB is strongly associated with climatic conditions and seasonal consumption habits. This result, related to 2024, confirms the seasonality observed through the time series analysis.

Analysis of search intent (Figure 6) indicates a predominance of transactional queries (71%), followed by informational queries (50%). The prevalence of transactional intent suggests that many users engage in searches that are likely to result in specific actions, including purchases, subscriptions, or downloads. The relatively high proportion of informational intent reflects a continuing interest in product knowledge, whereas the minimal navigational component indicates limited brand awareness within the category.

Table 2 presents the ten leading websites ranked by traffic share and ZA. ZA is a proprietary SEOZoom metric ranging from 0 to 100 on a logarithmic scale, incorporating ranking stability, growth potential, and Google’s perceived trustworthiness. Compared with other authority metrics (e.g., Moz, Ahrefs, SEMrush), ZA integrates visibility and traffic performance across domains.

The traffic distribution was relatively fragmented. The leading domain, birredamanicomio.com, captured 26.17% of the traffic with a moderate ZA of 52, indicating that topical relevance and optimised content may outweigh absolute authority. Bernabei.it (16.54%; ZA 60) and drinkami.shop (16.51%; ZA 48) also demonstrated a strong visibility. In contrast, Amazon.it—despite a ZA of 94—accounted for only 4.20% and 1.97% of traffic in two separate entries, suggesting that high-authority generalist platforms may underperform against niche sites in keyword-specific queries. The presence of health-oriented portals such as celiachia.it illustrates the importance of medical and dietary considerations in this search category.

Semantic mapping of user queries (Figure 7) offers further insight into search behaviour, classifying queries into three clusters: Questions, Prepositions, and Actions. The Prepositions map highlights a strong interest in ingredients (e.g., “birra senza glutine con lievito”—gluten-free beer with yeast), consistent with health-related concerns among consumers with coeliac disease. The Questions map shows the demand for practical, marketing-oriented information (e.g., “dove comprare birra senza glutine”; “quale birra è senza glutine”—where to buy gluten-free beer; which beer is gluten-free), reinforcing the transactional nature of searches. The Actions map focuses on procedural or purchase-driven queries (e.g., “come produrre birra senza glutine”—how to make gluten-free beer), suggesting that actionable information is prioritised over dietary or nutritional concerns in this subset of searches.

### 3.3. Time Series Analysis for Non-Alcoholic Beers and Low-Alcoholic Beers

Analysis of online search volumes for the keyword NABLAB indicates that the first signs of user interest emerged in 2006 (Figure 8). Initial activity was sporadic until 2010, after which searches increased consistently, reaching a peak in 2024. The results highlight a strong, long-term upward trend (green line), reflecting the growing diversification of beer consumption.

The time series decomposition revealed a distinct seasonal component, with recurrent peaks and troughs following a regular annual structure. A comparison of the fitted and observed values demonstrated satisfactory model adherence, with deviations confined to extreme peaks, likely reflecting exceptional events to be examined separately. Forecasts for 2025–2030 suggest continued growth in search volumes, maintaining the established seasonal structures.

Model accuracy measures confirmed reliability: a MAPE of 18.64% indicated moderate predictive error within acceptable limits; a MAD of 5.29 reflected relatively low mean deviations from observed values; an MSD of 58.30 suggested limited residual variability, with no substantial systematic errors. Overall, the multiplicative decomposition model effectively captured the underlying temporal dynamics, providing an interpretable representation of consumer interest and a robust basis for medium-term forecasts.

Seasonal analysis (Figure 9) confirmed systematic variations in demand. Search volumes peak during the summer months, particularly in July, when the seasonal index exceeds 1.5 (i.e., more than 50% above the annual average). In contrast, December and January show indices close to 0.6, indicating a contraction in demand in winter. The percentage change by season highlights July as an anomaly, potentially linked to external factors such as promotional campaigns, favourable weather, or specific market events. Seasonally adjusted data also show an extreme positive deviation in July, which is consistent with the extraordinary circumstances. Residual analysis indicated limited variability across most months, except in July, when the residuals were notably higher. Given the GT scale (0–100), the observed anomalies should be interpreted within these constraints. Overall, the findings confirm a well-defined annual seasonality, with summer peaks and winter declines, while emphasising the influence of contextual factors in the interpretation of anomalies.

### 3.4. SEOzoom Analysis for NABLAB

An additional analysis was conducted using the SEOZoom tool for the keyword NABLAB. Examination of the monthly variation in indexed SERPs revealed an average search volume of 4.4 k and approximately 486 k indexed pages. The dataset showed substantial fluctuations, ranging from 451 k to 644 k pages (Figure 10). Two pronounced peaks were observed, the first in April (575 k) and the second, higher, in November (644 k), interspersed with phases of stabilisation or decline.

This non-linear pattern may reflect changes in Google’s indexing behaviour, associated either with increased publishing activity or algorithmic prioritisation of content. These peaks are plausibly linked to marketing campaigns, regulatory developments, or heightened consumer attention.

Figure 11 illustrates the seasonal nature of user searches, expressed as the average monthly volume over a twelve-month period. As a proxy for digital demand, this metric highlight pronounced seasonality: searches rise sharply from May to August, peaking in June (8100), before declining and stabilising at around 2900 towards the year-end. Smaller peaks in May (4400) and September (4400) indicate anticipatory and residual interest in the summer period. In contrast, reduced volumes in colder months suggest lower consumption intent and limited promotional activity.

The search intent distribution (Figure 12) shows a predominance of transactional queries (64.0%), followed by informational (38.6%), commercial (11.0%), and minimal navigational intent (1.0%). Transactional dominance indicates a strong user propensity towards concrete actions, whereas informational intent confirms sustained interest in product attributes, health implications, and benefits. The smaller commercial component suggests exploratory engagement, and the low navigational intent indicates limited brand loyalty, consistent with an emerging market segment.

Table 3 presents the ten websites with the highest keyword traffic alongside their respective ZA scores. The traffic distribution was heterogeneous. The leading domain, www.zeroalcol.com, accounted for 40.64% of traffic despite a moderate ZA score (41), indicating strong optimisation and keyword targeting. In contrast, Amazon ranked fifth (5.58%) despite the highest ZA score (94), underscoring the relative advantage of niche platforms in keyword-specific visibility. This confirms previous observations for GFB, where domain authority alone does not guarantee SERP dominance.

Specialised domains such as www.myalcolzero.it (9.24%), www.shop.baladin.it (5.84%), and www.birramoretti.com (4.18%) further highlight the importance of targeted, product-specific platforms. Their mid-range ZA scores (34–54) suggest that keyword relevance and optimisation can outweigh general authority in driving organic traffic.

To complement these findings, Figure 13 presents semantic maps of NABLAB-related queries classified into interrogative forms (Questions), prepositional phrases (Prepositions), and verbal expressions (Actions).

The Questions map reflects interest in comparative evaluations and purchase-related information (e.g., “qual è la migliore birra analcolica”—what is the best alcohol-free beer). Persistent queries about points of purchase suggest that NABLAB products are not yet universally available in the market. Informational queries, such as “birra analcolica cosa è?” (what is alcohol-free beer?), highlight user demand for definitional clarity.

The Prepositions map underscores health-related concerns, with frequent queries on product composition and suitability (e.g., “birra analcolica senza glutine”—gluten-free non-alcoholic beer; “birra analcolica in gravidanza”—alcohol-free beer during pregnancy). These findings highlight consumer sensitivity to dietary restrictions and prenatal safety.

The Actions map reveals perceptual and behavioural concerns, including health implications (e.g., “fa ingrassare”—does it cause weight gain; “gonfia la pancia”—does it cause bloating) and legal aspects (“si può vendere ai minorenni”—can it be sold to minors). Unlike GFB, NABLAB queries emphasise health, regulatory, and lifestyle dimensions.

Taken together, these findings confirm that NABLAB remains in the discovery phase, with consumer interest shaped by both transactional intent and health-related concerns.

## 4. Discussion

This study proposed an integrated methodological framework that combines time-series decomposition with descriptive SEO-based analysis to investigate consumer interest in emerging beer categories. When applied to the Italian market, this approach provides new insights into two key product segments: GFB and NABLAB.

The discussion is structured around the study’s guiding research questions. 1. In relation to observed market changes concerning GFB and NABLAB, can the evolution of beer consumption be confirmed through the online search activity of consumers? (RQ1) 2. What new patterns of consumer interest and intent are emerging in the Italian beer market, and what future directions are likely to shape demand? (RQ2) 3. How can the integration of time-series analysis and descriptive SEO analysis offer a more robust framework for understanding consumer behaviour in niche food and beverage markets? (RQ3)

In addressing RQ1, the results of this study are particularly significant when viewed in the context of broader online interest in beer-related topics. Indeed, while overall search volumes for conventional beer have remained relatively stable over the years (in both the global and Italian markets), the data reveal a steady and measurable increase in online searches for GFB and NABLAB. This trend suggests that the observed growth in search activity is not simply a reflection of an overall increase in beer-related web queries, but rather the result of a genuine and autonomous increase in consumer interest in these specific product categories. The methodological choice to rely on targeted, category-specific keywords was crucial in isolating this phenomenon. By excluding generic beer-related search terms, the analysis minimises the risk of data contamination from trends associated with traditional beer consumption.

The time-series analysis of GT data revealed a clear upward trajectory in search interest over the past two decades, characterised by strong seasonality, with peaks in the summer months and troughs in winter. Forecasting indicates the persistence of this growth pattern until 2030, confirming (per RQ1) the structural rather than episodic nature of demand. This longitudinal evidence is crucial because it validates and reinforces the findings derived from SEOZoom. Whereas SEOZoom provides a granular view of search intent, semantic associations, and indexed content, time-series analysis demonstrates the durability and cyclicality of these patterns, transforming descriptive signals into evidence of stable behavioural trends.

The complementarity of the two tools is central to answering RQ3. SEOZoom excels at revealing what consumers search for and how they articulate their queries, but it lacks a temporal dimension. In contrast, Google Trends captures when and to what extent these searches occur [58], but without the semantic depth offered by SEOZoom. By integrating the two, we constructed a more comprehensive analytical framework: time-series analysis supplies the structural backbone of consumer behaviour, while SEOZoom adds interpretive depth regarding motivations, concerns, and digital pathways.

In relation to RQ2, the evidence of strong seasonal demand makes it strategic to intervene in winter months through data-driven marketing approaches. Analysis of queries related to the cold season allows to identify different needs and search intentions compared to summer, providing useful insights for the design of targeted campaigns. Mapping semantic patterns (questions, prepositions, actions) allows for the development of “winter-fit” campaigns that position these beers as valid alternatives to wine and spirits, which traditionally dominate indoor consumption in winter. Among the actions with the greatest potential are landing pages optimised for informational and transactional intentions. This approach facilitates a process of “deseasonalization” of demand, contributing to changing consumption patterns. This would reduce the amplitude of the supply-demand cycle and increase winter sales, with a view to achieving greater alignment between editorial planning, advertising, and trade promotion activities in relation to the interests expressed by users during the cold months.

Thus, this integrated approach generates actionable marketing implications. For GFB, the interplay between informational and transactional queries suggests the need for dual content architectures: product-oriented assets (e.g., store locators, e-commerce listings) alongside educational resources addressing health concerns. For NABLAB, the prevalence of searches related to legality, health effects, and situational consumption underscores the importance of authoritative, evidence-based content. Time-series analysis further indicates that these strategies should be strategically timed to coincide with seasonal peaks in summer when demand intensifies.

Further addressing the consumer patterns in RQ2, the predominance of transactional intent associated with the keywords suggests that a significant proportion of users conduct active searches, typically followed by a specific action. These actions may include making a purchase, subscribing to a newsletter, downloading informational content, or any activity involving a concrete outcome that is not necessarily economic. This finding underscores the commercial significance of the data in relation to user interests and its potential implications for marketing strategy. Moreover, a notable proportion of informational intent reflects users’ ongoing interest in acquiring knowledge about product categories. In contrast, the minimal level of navigational intent indicates that relatively few users search for a specific website or brand. This suggests that, at this stage of market development, brand awareness in this segment remains limited.

From a theoretical standpoint, this study advances the literature by demonstrating the robustness of the framework (RQ3). It shows that SEO performance in niche food and beverage markets is driven less by traditional metrics such as domain authority and more by semantic coherence, topical relevance, and the capacity to engage with overlapping user intent. This evidence confirms that websites with significantly lower domain authority than more structured platforms (such as Amazon) can still achieve greater visibility, particularly when targeting niche markets such as those under analysis, if supported by efficient semantic optimisation. This aligns with the growing body of research highlighting the importance of subject-matter authority in health- and nutrition-related domains [59,60]. The evidence thus shows that smaller platforms can compete effectively with larger actors by leveraging focused, intent-driven strategies supported by temporal analysis.

For manufacturers and retailers, these digital signals (addressing RQ2) enable them to segment consumers (e.g., medical-driven vs. wellness-driven users), calibrate promotional messages and portfolios (flavours, formats, claims), plan production and logistics more efficiently during seasonal peaks, and optimise media budget allocation between informational and transactional content. Furthermore, comparative evidence between GFB and NABLAB allows us to distinguish between the regulatory/clinical drivers of adoption (gluten-free diet) and voluntary/value-based drivers (alcohol reduction), clarifying how preferences and brand loyalty evolve in categories that are still consolidating. The proposed analysis offers a sensitive and timely lens through which to interpret the evolution of beer consumption behaviour and translate it into data-driven marketing decisions. The opportunity to leverage the evolution of the beer industry with the potential of digital marketing can prove to be a viable strategy for the resilience of small businesses in this evolving brewing sector [61,62,63,64].

The methodological implications (RQ3) are equally significant. While each tool has limitations in isolation (Google Trends provides relative rather than absolute data, and SEOZoom outputs are model-based estimates), together they deliver a synergistic framework capable of validating traditional market research while revealing emergent behavioural signals. The combination of temporal analysis and descriptive web analytics offers a cost-effective, scalable, and adaptable model that can be replicated in other markets and product categories in the future.

The adopted model can also be replicated in different geographical contexts. Time series analysis is based on standardised statistical procedures and can be applied to Google Trends data from any country. SEO analysis based on SEOZoom has an intrinsic domain limitation, as coverage is linked to the Italian domain; however, it is methodologically transferable using similar tools available in other markets (e.g., SERP analysis software for foreign domains with comparable metrics). Thus, the combination of the two approaches constitutes a generalizable framework for describing and predicting the evolution of demand in other countries. However, operational limitations remain, such as aligning language and local synonyms, considering GT normalisation for international comparisons, and calibrating models in the presence of cultural or seasonal differences. Overall, the proposed protocol is replicable, transparent, and comparable, providing a solid basis for cross-country studies and data-driven marketing decision-making.

## 5. Conclusions

In conclusion, this study successfully demonstrates that online search behaviour acts as a reliable mirror for emerging market dynamics, providing both substantive and methodological contributions. Substantively, the analysis confirms the growing consumer interest in GFB and NABLAB, which is shaped by distinct health concerns and lifestyle factors. The time-series decomposition of Google Trends data reveals a structural, long-term positive trend for both categories, a finding validated by the identification of strong, recurrent seasonality peaking in the summer months. The semantic analysis further reveals the emergence of two distinct consumer profiles: the interest in GFB is mature and need-driven, dominated by specific health concerns and transactional queries, whilst interest in NABLAB is more exploratory, with consumers still defining the category’s physiological and situational contexts. Crucially, the minimal navigational (brand-specific) search intent for both products suggests that these markets are still consolidating and brand loyalty is low, presenting a significant opportunity for market penetration. Methodologically, this study introduces an integrated model that leverages the strengths of time-series decomposition and descriptive SEO analysis. This methodological fusion is key to producing a richer, multidimensional understanding of digital consumer behaviour. The time-series analysis provides the “when” and “how much”, validating the trend’s durability and cyclicality, whilst the SEO analysis provides the “what” and “why,” deconstructing consumer motivations through semantic mapping and intent classification. While each tool has limitations in isolation—Google Trends provides relative data, and SEOZoom outputs are model-based estimates—together they deliver a synergistic framework capable of validating traditional market research whilst revealing emergent behavioural signals.

From a managerial and strategic perspective, these findings offer valuable implications for enterprises, especially for small and medium-sized (SMEs) in the Italian brewing sector. By leveraging real-time data on consumer search behaviour, breweries can monitor how preferences evolve and identify emerging trends before they fully materialise. This integrated framework moves beyond simple trend-spotting to offer actionable insights. It strategically explains why niche, content-driven domains outperform high-authority generalist platforms: they successfully answer the complex informational and transactional queries of a consumer base that requires education and trust before purchasing. This approach provides SMEs with accessible, cost-effective means to enhance their market intelligence, strengthen their competitive position, and contribute to sustainable innovation.

By reinforcing descriptive insights with temporal validation, this approach transforms fragmented digital signals into robust and interpretable evidence, offering value for both academic research and industry practice. Future research should therefore extend this integrated framework beyond Italy to test its applicability in cross-national contexts, analysing differences in seasonality, search intent, and semantic structures to uncover transnational patterns of consumer engagement.

## Figures and Tables

**Figure 1 foods-14-03789-f001:**
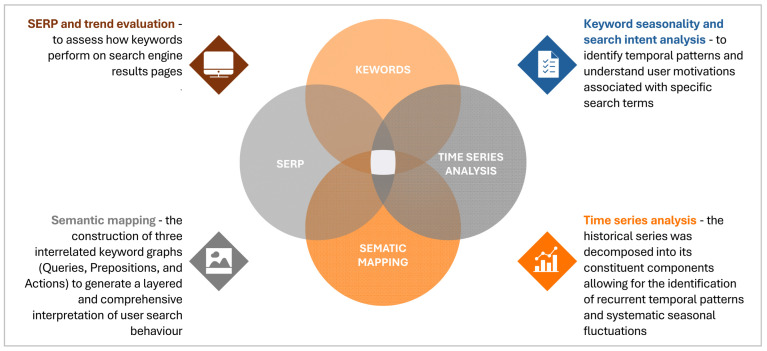
Methodological approach adopted.

**Figure 2 foods-14-03789-f002:**
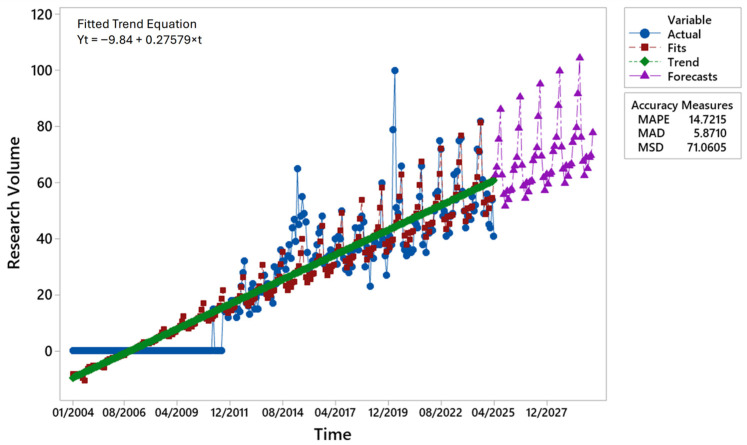
Time series decomposition of online searches for the GFB keyword using a multiplicative model (January 2004–April 2025) with a forecast to 2030. Author’s elaboration based on the GT data. Note: GT data were normalised on a scale of 0–100. Each point is expressed relative to the maximum value, and the aggregated values correspond to relative, not absolute, search volumes.

**Figure 3 foods-14-03789-f003:**
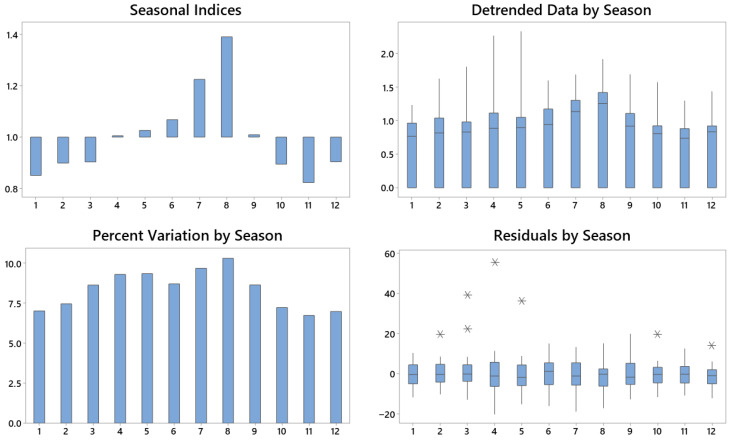
Seasonal decomposition of online searches for the GFB keyword. Author’s elaboration based on GT data.

**Figure 4 foods-14-03789-f004:**
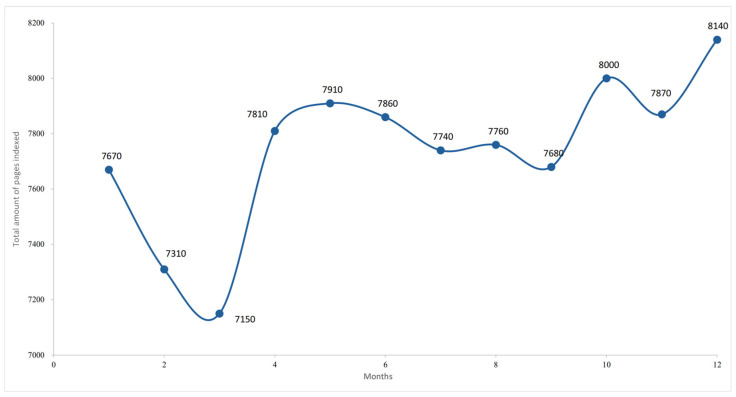
Monthly variation in indexed pages for the GFB keyword (‘000). The author’s elaboration is based on the Google SERP data.

**Figure 5 foods-14-03789-f005:**
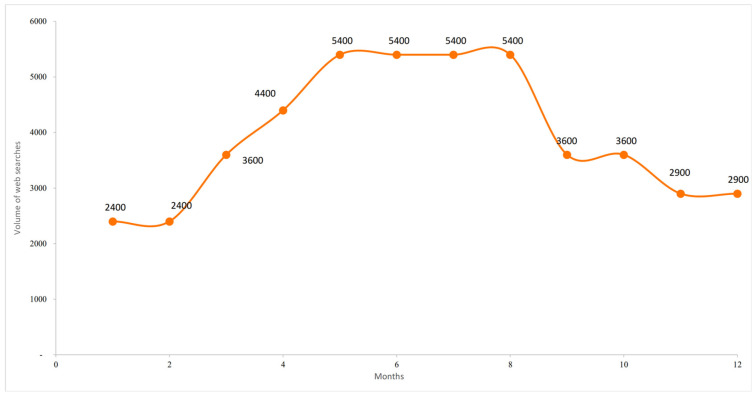
Seasonal distribution of searches for the GFB keyword. Author’s elaboration based on SEOZoom.

**Figure 6 foods-14-03789-f006:**
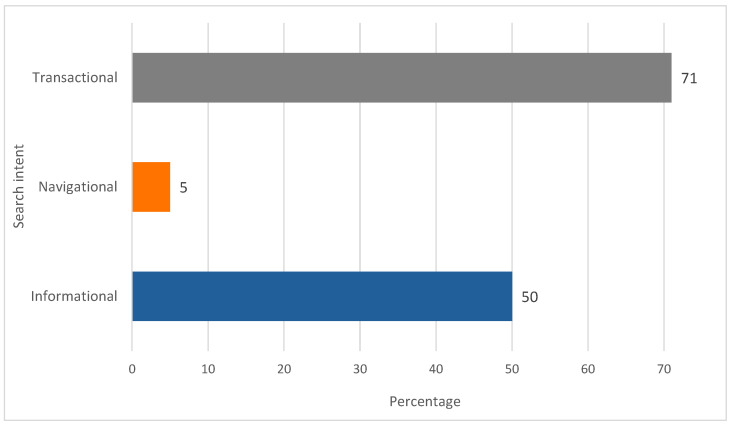
Search intent distribution for the GFB keyword (values in %). Author’s elaboration based on SEOzoom. Note: Percentages exceed 100% owing to overlap, as some queries satisfy multiple intent categories.

**Figure 7 foods-14-03789-f007:**
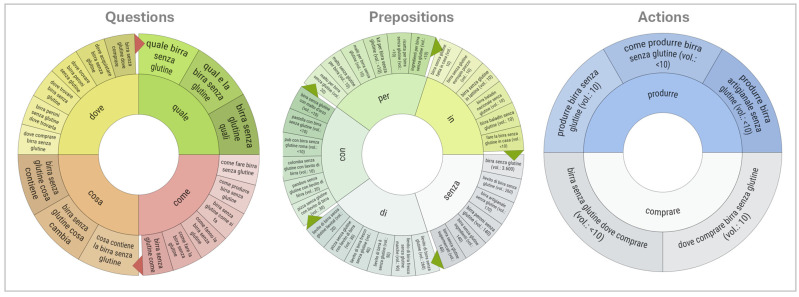
Semantic maps of queries for the GFB keyword. Author’s elaboration based on SEOzoom. Note: The English translation is available in the Appendix B.

**Figure 8 foods-14-03789-f008:**
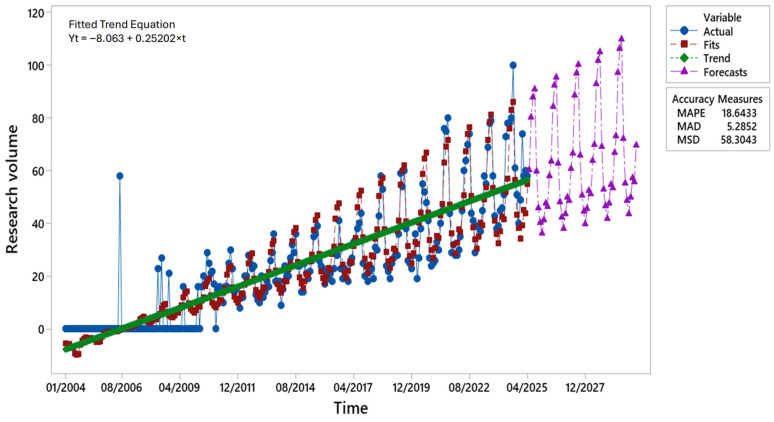
Time series decomposition of online searches for the keywords non-alcoholic and low-alcoholic beers using a multiplicative model (January 2004–April 2025) with a forecast to 2030. Author’s elaboration based on GT data. Note: GT data are normalised on a scale of 0–100. Each point is expressed relative to the maximum value, and aggregated values correspond to relative, not absolute, search volumes.

**Figure 9 foods-14-03789-f009:**
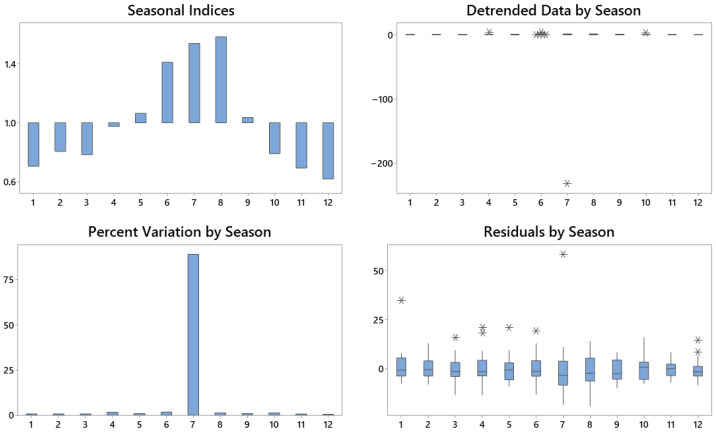
Seasonal decomposition of online searches for the NABLAB keyword. Author’s elaboration based on GT data.

**Figure 10 foods-14-03789-f010:**
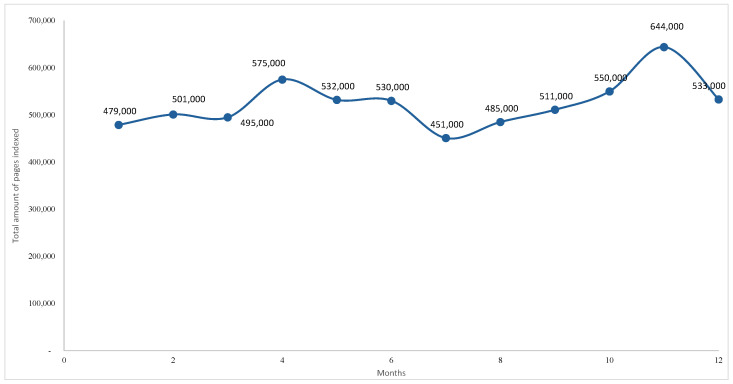
Monthly variation in indexed pages for the NABLAB keyword. Author’s elaboration based on Google SERP data.

**Figure 11 foods-14-03789-f011:**
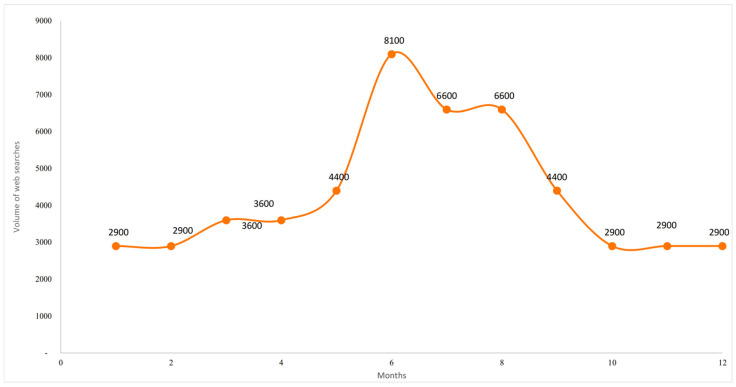
Seasonal distribution of searches for the NABLAB keyword. Author’s elaboration based on SEOzoom.

**Figure 12 foods-14-03789-f012:**
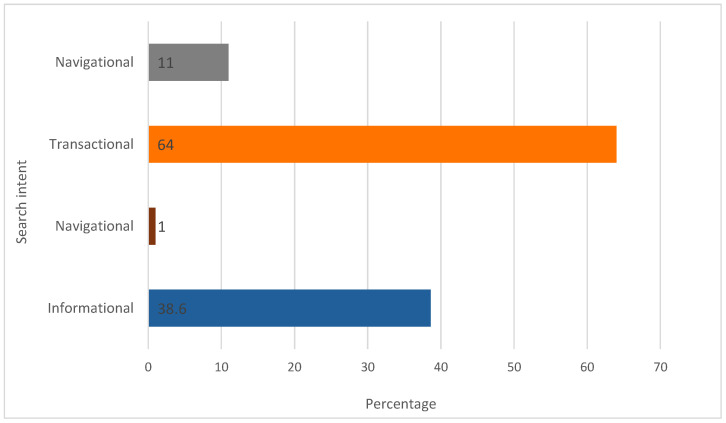
Search intent distribution for the NABLAB keyword (values in %). Author’s elaboration based on SEOzoom. Note: Percentages exceed 100% owing to overlap, as some queries satisfy multiple intent categories.

**Figure 13 foods-14-03789-f013:**
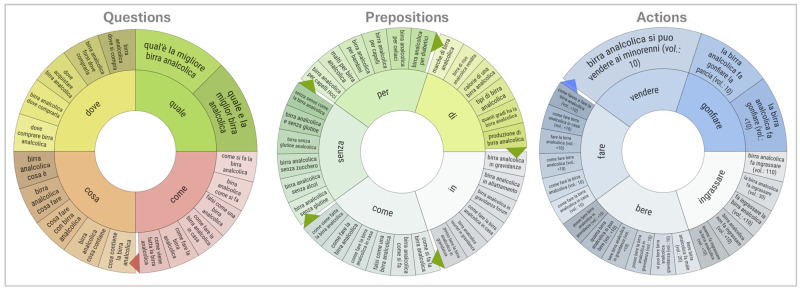
Semantic maps of queries for the NABLAB keyword. Author’s elaboration based on SEOZoom. Note: An English translation is provided in the Appendix B.

**Table 1 foods-14-03789-t001:** Description of SEOZoom metrics used in the analysis. Author’s elaboration.

SEOZoom Indicators	Description
Variation in indexed pages	This metric quantifies the monthly change in the number of web pages indexed by search engines that contain the selected keywords. It provides an estimate of online content production and competitive density, offering insight into the dynamism of the digital information supply.
Variation in search volumes	SEOZoom provides an estimate of the monthly frequency with which a keyword is searched on Google. This parameter, analogous to GT yet expressed in absolute values, measures user demand and its temporal fluctuations, thereby reflecting interest dynamics and seasonality.
Search intent classification	Each keyword is associated with a dominant search intent (e.g., informational, navigational, transactional, or mixed). This classification helps to identify whether users are seeking knowledge, specific brands, or purchase opportunities, thus providing insight into the consumer’s decision-making stage.
Traffic share per domain	This indicator estimates the percentage of total organic traffic captured by a specific website from a given keyword cluster. It reveals the competitive positioning of different domains and the concentration of user attention within that online ecosystem.
Zoom Authority (ZA)	A proprietary SEOZoom metric, the ZA score quantifies a website’s authority and competitiveness. It is derived from multiple factors, including organic traffic, backlink profile, and keyword performance. Higher ZA values indicate stronger online visibility and perceived reliability. The indicator is expressed as a scaled value from 0 to 100, where higher scores reflect greater credibility and SEO authority as assessed by the platform.
Semantic maps	SEOZoom’s semantic analysis tool identifies the network of related keywords and topics surrounding a main query. This feature facilitates the exploration of semantic proximity, the identification of thematic clusters, and the detection of emerging trends in user search behaviour.

**Table 2 foods-14-03789-t002:** Top 10 websites by traffic share and Zoom Authority scores for the GFB keyword. Author’s elaboration based on SEOZoom data.

	URL	Traffic Share (%)	ZA * Scores
1	https://www.birredamanicomio.com/birre/senza-glutine/	26.17	52
2	https://www.bernabei.it/birre-gluten-free	16.54	60
3	https://www.drinkami.shop/stile/gluten-free/	16.51	48
4	https://www.1001.it/birra-senza-glutine	9.26	49
5	https://shop.baladin.it/products/nazionale-gluten-free	8.36	40
6	https://www.amazon.it/birra-glutine/s?k=birra+senza+glutine	4.20	94
7	https://www.cantinadellabirra.it/shop/degustazione-birre-senza-glutine.html	3.19	56
8	https://www.cantinadellabirra.it/shop/blog/stile/birre-senza-glutine.html	3.18	56
9	https://www.celiachia.it/faq/7-birra-e-bevande-a-base-di-orzo-o-frumento-dichiarate-senza-glutine-sono-sicure-per-i-celiaci/	2.25	54
10	https://www.amazon.it/birre-glutine/s?k=birre+senza+glutine	1.97	94

* Zoom Authority (ZA). Higher scores reflecting greater credibility.

**Table 3 foods-14-03789-t003:** Top 10 websites by traffic share and Zoom Authority scores for the NABLAB keyword. Author’s elaboration based on SEOzoon.

	URL	Traffic Share (%)	ZA * Scores
1	www.zeroalcol.com	40.64	41
2	www.drinkshoponline.com	15.56	54
3	www.myalcolzero.it	9.24	37
4	www.shop.baladin.it	5.84	41
5	www.amazon.it	5.58	94
6	www.birramoretti.com	4.18	34
7	www.birredamanicomio.com	3.96	52
8	www.birrificiodelducato.it	2.37	41
9	www.qualitybeeracademy.it	2.02	41
10	www.carrefour.it	1.96	67

* Zoom Authority (ZA). Higher scores reflecting greater credibility.

## Data Availability

The raw data supporting the conclusions of this article will be made available by the authors on request.

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
