# Peer review of "Digital Trends in the Italian Beer Market: A Time-Series and Search Engine Optimisation Analysis of Gluten-Free and Low/No-Alcohol Beers"

_foods, 2025, doi:10.3390/foods14213789_

Round 1

Reviewer 1 Report

Comments and Suggestions for Authors

To examine whether online search behaviour mirrors these steady growth of gluten-free beers (GFB) and low- and no-alcohol beers (NABLAB), this study combined time-series analysis of Google Trends data and SEOZoom platform, and showed that a consistent rise in online interest for both GFB and NABLAB with different thematic orientations and predicted the development trend for the future five years. These results provide a good reference for producers and consumers. However, there are some issues that need to be revised.

  1. Figure 1: The keywords used in the search are very important, but they are not clearly reflected in this figure, nor are the key analysis methods presented. It is recommended to supplement detailed key information to make it easier for readers to grasp the key points.
  2. “2.3. SEOzoom tool”: In this platform, the key software used should be clearly indicated, such as website, Zoom Authority (ZA), etc. Detailed usage methods should be supplemented.
  3. In Figures 4 and 5, based on Google SERP data and SEOZoom respectively, the authors analyzed the seasonal distribution of searches for the keyword gluten-free beer. Why are the highest points obtained inconsistent? Should the author explain or discuss this phenomenon? The same problem is also in Figures 10 and 11.
  4. Based on the existing search results, it is recommended to supplement more prediction results to provide references for producers and researchers.
  5. The current conclusions cannot summarize the entire text, it should include the main conclusions obtained through this research.
  6. The abstract does not fully reflect the results of this manuscript, such as the predicted future trends, etc.
  7. In the abstract and main text, the abbreviations GFB and NABLAB and their full names are used alternately. It is recommended to use the full names and their abbreviations when they first appear in the abstract and the main text, and then use the abbreviations instead.
  8. All the pictures should be drawn more standardly and in line with the format of the journal. For instance, the descriptions of the vertical and horizontal coordinates and their units in many pictures are missing, which should be completed. Figures 5 and 11 lack vertical coordinates, while Figures 7 and 12 lack horizontal coordinates. The month data used in figures 3-5 and others should indicate the specific year, and their expression of month in numbers or words should be consistent.
  9. Table 1: The number of decimal places should be consistent.
  10. Table 2: "%" is indicated in the first row, so there is no need to write "%" in each of the following rows, and it should be removed.
  11. Figures 7 and 13: Many words in these figures are not clear. It is suggested that they be condensed and presented in the figures. The corresponding brief descriptions should be noted in the text descriptions of the figures.
  12. It is suggested that the Discussion and conclusions be written separately. The suggested conclusions section should clearly state the main research conclusions, limitations and prospects of this article.
  13. References should be cited in a standardized manner. The citation format in places like L171/235 should be checked in full text.

Author Response

Comments 1: Figure 1: The keywords used in the search are very important, but they are not clearly reflected in this figure, nor are the key analysis methods presented. It is recommended to supplement detailed key information to make it easier for readers to grasp the key points.

Response 1: We appreciate your feedback and offer the following clarification regarding our manuscript's structure. Figure 1 outlines the methodological approach adopted for this research. We have indeed included keyword analysis as a critical component of our methodology. However, we classified the specific keywords identified through this process as results. This decision was based on the fact that they were derived from extensive iterative testing on Google Trends, rather than being pre-selected. Therefore, these keywords are presented as an initial finding in the first paragraph of the Results section. We trust this clarification adequately explains our rationale.

Comments 2: “2.3. SEOzoom tool”: In this platform, the key software used should be clearly indicated, such as website, Zoom Authority (ZA), etc. Detailed usage methods should be supplemented.

Response 2: We appreciate your suggestion, which has helped us improve the clarity of the manuscript.

Accordingly, Section 2.3 (‘SEOZoom Tool’) has been updated to provide a more detailed and clear description of the specific SEOZoom metrics used. Furthermore, a new table (Table 1) has been added to this section to explicitly summarize these methodological details. We hope this revision adequately addresses your comment.

Comments 3: In Figures 4 and 5, based on Google SERP data and SEOZoom respectively, the authors analyzed the seasonal distribution of searches for the keyword gluten-free beer. Why are the highest points obtained inconsistent? Should the author explain or discuss this phenomenon? The same problem is also in Figures 10 and 11.

Response 3: Thank you for this constructive feedback. We offer the following clarifications.

Regarding the apparent inconsistency in Figures 4, 5, 10 and 11.

The figures are not contradictory as they measure different metrics. We have revised the manuscript to explain this more clearly.

Figures 4 and 10 (Indexed Pages): This chart tracks the number of pages indexed by Google. This metric shows a general growth over time, reflecting an increase in the "supply" of content on this topic.

Figure 5 and 11 (Search Volume): This chart tracks the volume of user web searches. This metric reflects user "demand" and correctly follows the seasonal trend discussed.

Both data sets are obtained via SEOZoom, which analyzes Google SERP data for both indexed pages and user search estimates.

This comparison reinforces our findings. Important clarification is important to distinguish data types:

SEOZoom: Provides estimated absolute search volumes.

Google Trends: Provides relative indices (0-100).

The data reported by SEOzoom in reference to online searches conducted by users for a given keyword are different in nature from those reported by Google Trends. The fact that both tools, with their different methodologies, confirm the same strong seasonality for user searches validates the observed phenomenon.

Comments 4: Based on the existing search results, it is recommended to supplement more prediction results to provide references for producers and researchers.

Response 4: Thank you for this valuable suggestion. We agree that integrating forecast results would significantly enhance the paper's implications for manufacturers and researchers.

However, based on our comprehensive review of the current literature, we did not find any similar studies or established models that could provide a reliable foundation for such forecasts. Given the novelty of this specific research, any forecast results would be speculative at this stage. We have noted this as an important direction for future research

Comments 5: The current conclusions cannot summarize the entire text, it should include the main conclusions obtained through this research.

Response 5: Thank you for this important feedback. We have revised the Conclusions section to address your concern.

The new version has been rewritten to be more comprehensive and to better synthesize the main findings obtained from our research."

Comments 6: The abstract does not fully reflect the results of this manuscript, such as the predicted future trends, etc.

Response 6: Thank you for your valuable comment. We have revised the abstract to include more specific and detailed information, ensuring it now provides a completer and more explanatory summary of our main findings.

Comments 7: In the abstract and main text, the abbreviations GFB and NABLAB and their full names are used alternately. It is recommended to use the full names and their abbreviations when they first appear in the abstract and the main text, and then use the abbreviations instead.

Response 7: Thank you for your valuable comment. We have revised the text to correct this, ensuring that we no longer alternate between full names and abbreviations and now follow the standard convention you suggested.

Comments 8: All the pictures should be drawn more standardly and in line with the format of the journal. For instance, the descriptions of the vertical and horizontal coordinates and their units in many pictures are missing, which should be completed. Figures 5 and 11 lack vertical coordinates, while Figures 7 and 12 lack horizontal coordinates. The month data used in figures 3-5 and others should indicate the specific year, and their expression of month in numbers or words should be consistent.

Response 8: Thank you for your valuable and detailed comment.

We have revised all figures to address your concerns. Specifically, we have added the missing vertical and horizontal coordinate labels and units to all figures, including Figures 5, 7, 11, and 12.

Regarding the specific year for Figures 3-5, we would like to clarify that this is explicitly stated in Section 2.3, which defines the analysis period for the entire study (“The analysis covered the period from January 1, 2024, to December 31, 2024...”). To avoid redundancy, we had not repeated this information in every figure caption, as the context is established in the methodology. We hope this clarification is satisfactory.

Comments 9: Table 1: The number of decimal places should be consistent.

Response 9: Thank you for catching that. We have corrected the decimal places in Table 1 (now Table 2) for consistency

Comments 10: Table 2: "%" is indicated in the first row, so there is no need to write "%" in each of the following rows, and it should be removed.

Response 10: Thank you for your valuable comment. All “%” symbols have been removed.

Comments 11: Figures 7 and 13: Many words in these figures are not clear. It is suggested that they be condensed and presented in the figures. The corresponding brief descriptions should be noted in the text descriptions of the figures.

Response 11: Thank you for your valuable comment.

We acknowledge that some words in the figures are not clearly legible. The figures present semantic maps intended to visually illustrate the structure and treatment of semantic links. Since the analysis was conducted on Italian web traffic, the original content appears in Italian. To facilitate comprehension, we have included English translations of the key phrases corresponding to the most significant findings within the text. Furthermore, Tables A1 and A2 in the Appendix provide complete translations of semantic maps to ensure greater clarity and understanding.

We hope that this approach will make the content of the semantic maps accessible and clear to the reader.

Comments 12: It is suggested that the Discussion and conclusions be written separately. The suggested conclusions section should clearly state the main research conclusions, limitations and prospects of this article.

Response 12: Thank you for your valuable comment.

We have revised the entire paragraph. The discussions and conclusions are now presented in two separate paragraphs, which we have revised and supplemented. We hope that we have improved this section and addressed the issues that were raised.

Thank you for this valuable suggestion. We have revised this final part of the manuscript as requested. The Discussion and Conclusions are now presented in two separate sections.

Comments 13: References should be cited in a standardized manner. The citation format in places like L171/235 should be checked in full text.

Response 13: Thank you for your valuable comment. We have revised the citation format and adapted it to the journal’s standards.

Reviewer 2 Report

Comments and Suggestions for Authors
  1. 13-14 The beer do not show a growth. The consumption or the sales of these beers have.
  2. 106 You say that it contributes on the Italian market, but nothing in the title suggests that; the title rather indicated more global perspective. Consider modifying title of the manuscript.
  3. 254-255 Why so short period? Wouldn’t that period indicate mainly seasonal changes?
  4. 310-311 This correlates with beer in general. Is it therefore substantial for the gluten free beer, or just beer?

Figure 4 – is the x-axis months, with 1 being January, or not?

  1. 330-335 Once again – it probably correlates with beer in general.

The main problems of the article in general and in discussion section of this article is as follows:

  • Isn’t greater number of searches during the years correlating with general greater number of ALL searches worldwide?
  • Doesn’t the interest searches of gluten-free beers and low-alcoholic beers correlate with greater search of beer in general? So, is it really upward trend in searches, or these searches are still, let’s say, 2% of all beer-related search in 2010 and 2025, but in 2025 there was just more beer-related searches?
  • How does the so-called beer revolution, which resulted in worldwide spike in amount of breweries, interest in beer production and consumption, relate to these searches?

Without these, the article, in my opinion, is really speculative and might omit crucial factors influencing used data. All of these should be discussed thoroughly.

Author Response

Comments 1: 13-14 The beer do not show a growth. The consumption or the sales of these beers have.

Response 1: Thank you for your valuable comment. We have revised the abstract accordingly.

Comments 2: 106 You say that it contributes on the Italian market, but nothing in the title suggests that; the title rather indicated more global perspective. Consider modifying title of the manuscript.

Response 2: Thank you for your valuable feedback. We have revised the title to more accurately reflect the manuscript's focus on the Italian market.

Comments 3: 254-255 Why so short period? Wouldn’t that period indicate mainly seasonal changes?

Response 3: Thank you for you comment. We acknowledge the observation regarding the limited period used for the SEOZoom data analysis. The one-year period was intentionally selected because SEOZoom incorporates several dynamic indicators, such as intent analysis, Zoom Authority, semantic maps, that are optimally designed to capture short- to medium-term fluctuations within a one-year analytical window. In relation to temporal dynamics and seasonality, the SEOZoom analysis serves as an additional, short-term validation layer that supports and confirms the seasonal and structural trends detected through time-series analysis based on Google Trends data, covering the period of 21 years, from January 2004 to April 2025. Moreover, some SEOZoom indicators, such as Zoom Authority and semantic analysis, are computed dynamically and are not suitable for analysis over longer time spans, making the one-year period the most technically appropriate choice for this specific dataset, especially for semantic analysis that is related to more recent trends.

Comments 4: 310-311 This correlates with beer in general. Is it therefore substantial for the gluten free beer, or just beer?

Response 4: Thank you for this valuable observation. This result is indeed substantial for gluten-free beer.

The key finding is that, despite being a specialty product, its consumption follows the same seasonal patterns as the general beer market. One might have assumed that a niche “health-focused” product would have a different, less seasonal consumption cycle. Our finding confirms this is not the case. This has significant implications for marketing, as it demonstrates that strategies for gluten-free beer must account for this strong seasonality, just as they would for conventional beer. We have clarified the importance of this point in the discussion section.

Comments 5: Figure 4 – is the x-axis months, with 1 being January, or not?

Response 5: Thank you for pointing this out. Yes, the x-axis represents months (1=January). We have now revised Figure 4, and all other figures, to include all missing axis labels for clarity.

Comments 6: 330-335 Once again – it probably correlates with beer in general.

Response 6: Thank you for your valuable comment. The results presented in this study pertain specifically to gluten-free beers and hold considerable relevance for researchers, producers, and other stakeholders within the brewing sector. In particular, the findings are noteworthy as they confirm the seasonality of online searches for gluten-free beers using data that differ from those employed in the time series analysis. This constitutes an additional layer of validation that reinforces both the robustness of the applied methodology and the reliability of the results obtained.

It is important to underscore that all reported and discussed findings are derived from the analysis of data concerning gluten-free beers and beers with low or no alcohol content. All datasets were obtained through analyses based on the keywords identified in relation to these specific categories. Consequently, the results presented herein are strictly confined to these product niches and should not be extrapolated to beer in general.

Comments 7: The main problems of the article in general and in discussion section of this article is as follows:

Isn’t greater number of searches during the years correlating with general greater number of ALL searches worldwide?

Doesn’t the interest searches of gluten-free beers and low-alcoholic beers correlate with greater search of beer in general? So, is it really upward trend in searches, or these searches are still, let’s say, 2% of all beer-related search in 2010 and 2025, but in 2025 there was just more beer-related searches?

Response 7: Thank you for raising this critical point. The following answer addresses both questions.

The observed growth is not an artifact of general search increases but reflects a genuine, specific trend. This is for three main reasons:

Google Trends Data is Normalized: The data used in this study does not show absolute search volume. It is normalized, representing the relative search interest for a term compared to all other searches. Therefore, a rising trend line confirms that public interest in these products is growing faster than the general growth of global web searches.

Keyword Specificity: Our analysis used only specialized keywords (e.g., “birra senza glutine” – “gluten-free beer”). These terms do not overlap with broader "beer" searches, ensuring the observed trends are exclusively related to these niche categories.

Divergent Market Trends: This is the most direct answer to your question. When we compare our results with historical search data for the general term "beer," we find that searches for "beer" have remained relatively stable. In contrast, interest in GFB and NABLAB has increased sharply. This confirms a specific market shift toward these products, not a general rise in all beer-related searches."

Search results for the keyword “birra” for Italian market.

Search results for the keyword “beer” worldwide.

Comments 8: How does the so-called beer revolution, which resulted in worldwide spike in amount of breweries, interest in beer production and consumption, relate to these searches?

Response 8: We thank the reviewer for this insightful question. The so-called “beer revolution” is primarily associated with the global proliferation of craft breweries and the expansion of microbrewing culture. This movement has been largely driven by small and medium-sized enterprises (SMEs) that prioritize product differentiation, quality, and innovation over large-scale industrial production. Nevertheless, these smaller breweries often operate under substantial competitive pressure from major beer corporations, as they generally lack comparable economies of scale and cost efficiencies. Consequently, the ability to identify emerging consumer preferences in real time becomes essential for their survival and competitiveness. Within this context, our findings are particularly relevant. The observed increase in online interest toward gluten-free and low-/no-alcohol beers provides microbreweries and SMEs with valuable insights into evolving consumption patterns. Understanding these search dynamics enables producers to align their production strategies and marketing approaches with consumers’ growing emphasis on health, wellness, and responsible drinking. Therefore, the relationship between the “beer revolution” and the results of this study lies in the strategic use of digital intelligence to foster innovation, adaptability, and market responsiveness in a highly competitive and rapidly evolving brewing landscape.

Comments 9: Without these, the article, in my opinion, is really speculative and might omit crucial factors influencing used data. All of these should be discussed thoroughly.

Response 9: Thank you for this valuable comment. We agree that these considerations are essential for ensuring a rigorous and non-speculative analysis.

Accordingly, we have substantially revised the Discussion and Conclusions sections to address this point. In the revised version, we explicitly clarify why our results cannot be attributed to generic search behaviour or overall beer-related trends, emphasizing the role of data normalization and the divergence observed in the corresponding trend lines.

Round 2

Reviewer 2 Report

Comments and Suggestions for Authors

The manuscript was adequately corrected.